# Assessment of Post-Exertional Malaise (PEM) in Patients with Myalgic Encephalomyelitis (ME) and Chronic Fatigue Syndrome (CFS): A Patient-Driven Survey

**DOI:** 10.3390/diagnostics9010026

**Published:** 2019-03-02

**Authors:** Carly S. Holtzman, Shaun Bhatia, Joseph Cotler, Leonard A. Jason

**Affiliations:** Center for Community Research, Department of Psychology, DePaul University, Chicago, IL 60604, USA; choltzm1@depaul.edu (C.S.H.); sbhatia3@depaul.edu (S.B.); jcotler@depaul.edu (J.C.)

**Keywords:** myalgic encephalomyelitis, chronic fatigue syndrome, post-exertional malaise, assessment, patient-driven questionnaire, participatory research

## Abstract

Considerable controversy has existed with efforts to assess post-exertional malaise (PEM), which is one of the defining features of myalgic encephalomyelitis (ME) and chronic fatigue syndrome (CFS). While a number of self-report questionnaires have been developed to assess this symptom, none have been comprehensive, and a recent federal government report has recommended the development of a new PEM measure. The current study involved a community-based participatory research process in an effort to develop a comprehensive PEM instrument, with critical patient input shaping the item selection and overall design of the tool. A survey was ultimately developed and was subsequently completed by 1534 members of the patient community. The findings of this survey suggest that there are key domains of this symptom, including triggers, symptom onset, and duration, which have often not been comprehensively assessed in a previous PEM instrument. This study indicates that there are unique benefits that can be derived from patients collaborating with researchers in the measurement of key symptoms defining ME and CFS.

## 1. Introduction

Among patients with myalgic encephalomyelitis (ME) and chronic fatigue syndrome (CFS), post-exertional malaise (PEM) has long been considered a hallmark symptom [1]. However, in a field which includes more than twenty case definitions for ME and CFS, there has not been agreement regarding defining PEM [2]. For example, discrepancies occur with two of the most frequently used ME and CFS case definitions, the Fukuda [3] and Canadian Consensus Criteria (CCC; [4]). The Fukuda et al. criteria do not define the term beyond requiring that it last for more than 24 h nor does it make PEM a requirement for diagnosis. In contrast, the CCC case definition requires the presence of PEM for diagnosis and goes further to describe the symptomatic experience as similar to flu-like distress, with a potential delayed onset [4].

Several activity and self-report measurements that assess the extent of activity and how such activity might result in exacerbation of symptoms have been proposed to measure PEM. These include actigraphy, exercise challenges, time logs, and self-reports [5]. For example, following an exercise task, Mateo et al. [6] reported a broad spectrum of PEM-related symptoms including fatigue, muscle/joint pain, cognitive dysfunction, decrease in function, headaches, sleep disturbances, pain, weakness, cardiopulmonary symptoms, lightheadedness, and flu-like symptoms. Others have found using self-report measures that PEM comprises two distinct constructs: muscle-specific fatigue and generalized fatigue [7].

Factors which elicit PEM include physical and cognitive exertion. For some patients, even basic activities of daily living such as toileting, bathing, dressing, communicating, and reading can trigger PEM. However, many patients feel that potential triggers should extend beyond these types of stressors and include infections [8], exposure to chemicals or certain foods [9], or exposure to certain metals [10]. Additionally, many efforts to assess PEM have not included a characteristic delay in the onset of PEM. Chu et al. [11] maintain that this delay is rarely found in other fatiguing illnesses. Another issue that has often not been included in the assessment of PEM is that many patients with ME and CFS take considerably longer to recover from a trigger [12], reporting a substantial increase in symptoms immediately after an exercise test, the next day, and even a week later [13].

In an effort to address these PEM-related discrepancies, the National Institutes of Health/Center for Disease Control and Prevention (NIH/CDC) Common Data Element (CDE) committee’s PEM working group attempted to define PEM [14] as “an abnormal response to minimal amounts of physical or cognitive exertion that is characterized by: (1) Exacerbation of some or all of an individual study participant’s ME/CFS symptoms. (2) Loss of stamina and/or functional capacity. (3) An onset that can be immediate or delayed after the exertional stimulus by hours, days, or even longer. (4) A prolonged, unpredictable recovery period that may last days, weeks, or even months. (5) Severity and duration of symptoms that is often out-of-proportion to the type, intensity, frequency, and/or duration of the exertion.” Yet, there was no set of items with anchor points associated with these 5 descriptors of PEM offered by the NIH/CDC CDE PEM working group. While the general guidance of the committee was helpful, these types of general descriptions need to be operationalized if investigators are to reliably use them to assess PEM. The NIH/CDC CDE’s PEM working group also recommended the use of 5 items from the DePaul Symptom Questionnaire (DSQ, [12]) to measure PEM (e.g., physically drained or sick after mild activity). However, the DSQ was not developed as a comprehensive measure of PEM but rather as a measure of ME and CFS symptomatology as a whole. Following the release of the NIH/CDC CDE’s PEM recommendations, patients were extremely concerned with the recommendations that had been made [15].

This latest NIH/CDC CDE’s recommendations regarding the measurement of PEM needs to be understood in the context of a long history where patients have felt left out of key policy decisions imposed on them, including how to name, define, and treat ME and CFS. As one example, when the Institute of Medicine (IOM; [16]) recommended a new name and case definition, this created considerable controversy, as many feel that both were decisions imposed on the patient community, without first seeking their input and approval.

The recent recommendations made by this NIH/CDC CDE’s PEM working group, and the vociferous reactions to it by the patient community, provided an opportunity to engage in community-based participatory research, which equitably involves all partners in the research process [17]. Given the importance of PEM, and the patient community’s resentment regarding once again not being active participants in the development of this latest PEM recommendation, the current authors decided to try to develop a comprehensive measure with active collaboration of the patient community. We hypothesized that a valid PEM instrument could be created with the help of the patient community. 

## 2. Materials and Methods

### 2.1. Methods and Participants

The study began with dialogue between Leonard Jason and a number of leading patient activists who were unhappy with the NIH/CDC CDE’s PEM recommendations. A patient poll had indicated that the patient community preferred the NIH/CDC general description of PEM rather than the 5 DSQ items [15], but those general PEM descriptors had not been operationalized in any systematic way. Jason and several patient activists reworked those descriptors into a usable questionnaire, and this was posted on Facebook, Twitter, and LinkedIn social media pages and were widely shared with patient groups internationally. Hundreds of emails were received during the next three months, and Jason and Holtzman posted nine revisions of the survey for patients to provide comments. The comments and items received helped shape each new revision of the questionnaire. For example, when one participant commented “I also experience different types of PEM. I have the immediate PEM, where I do too much… But if I stop [exerting myself], these [PEM symptoms] go away fairly quickly… But if I am not able to stop during this immediate PEM stage and have to push on while experiencing these symptoms, then I get the “Post-PEM” usually two or more days later,” we used this input to introduce survey items that asked about both the immediate and delayed onset of PEM and its relationship to potential triggers.

After several months, when we were receiving few additional patient comments regarding our survey that we had been posting, we decided to collect data using this survey with the next phase of this project. Institutional Review Board (IRB) approval was obtained for collecting data based on the survey that had been developed using input from the patient community. Participants provided informed consent. Participants were required to be over the age of 18 years old, able to read and write in English, and have a current self-reported diagnosis of ME and/or CFS. Participants completed the questionnaire online using Research Electronic Data Capture (REDCap), a secure online survey tool [18]. Respondents were instructed to save their answers and return to complete the survey at a later time if they were not able to finish the survey in one sitting due to their illness.

### 2.2. Materials

The first part of the survey assessed demographic characteristics, as well as information about illness/diagnosis status (see Table 1). Following this background assessment, the respondents were asked about the onset of their PEM symptoms (see Table 2), and then asked questions relating to factors that trigger PEM (Table 3). This included examples of triggers beyond physical or cognitive exertion, such as “basic activities of daily living”, “positional changes”, and “emotional events”. The survey also asked specific questions about the relationship between triggers of PEM and other factors, such as participants’ individual energy limits or the extent to which they may exert themselves.

Next, the participants were asked to evaluate a list of symptoms that are exacerbated following physical and/or cognitive exertion (Table 4). The symptoms included items which have been assessed through other operationalized measures (e.g., “physical fatigue”, “unrefreshing sleep”, and “flu-like symptoms”), as well as items suggested by patients (e.g., “physical fatigue while mentally wired”, “brain twangs” and “burning sensation all over your skin”). Each item was rated for frequency for the past six months on a 5-point Likert scale: 0 = none of the time, 1 = a little of the time, 2 = about half the time, 3 = most of the time, 4 = all of the time. Symptoms of 2 or higher were considered to be the threshold for PEM, based on past studies [19]. For each symptom, frequency values were multiplied by 25 to convert to 100-point scales, with higher values indicating more frequent symptoms.

Table 5 shows item responses of participant experiences of PEM by asking the question “If you go beyond your energy limits by engaging in pre-illness tolerated exercise or activities of daily living, do you experience any of the following?” Several common phrases used to describe PEM were then listed, including “a severity and duration of symptoms that are out of proportion to the initial trigger” and “global worsening of multi-systemic symptoms (an example of this might be aches all over your body plus cognitive problems plus light and/or sound sensitivity)”.

Following the PEM symptom list, the survey included an assessment of duration of PEM and length of recovery time, as well as information about illness course and functioning (Table 6). To better understand the relationship between PEM and exertion, participants were asked if the severity and duration of PEM was out-of-proportion to the type, intensity, frequency, and duration of exertion. Participants were then asked whether they had ever experienced an “adrenaline surge” after going beyond their energy limit, and how long the surge lasts before the onset of PEM. Next, patients were assessed on their illness course and functional status by asking how long ago they began feeling sick with ME or CFS, if the illness has been present for at least 50% of the time, and how they would describe their illness and functioning. Participants were also asked if they are managing their PEM symptoms by pacing or “staying within their energy envelope,” one of the few patient recommended treatments for ME and CFS [20].

The survey also requested information about past tests the participant may have completed, such as a cardiopulmonary or tilt table test. Lastly, the survey assessed if the participants felt that this patient-driven survey accurately depicts their PEM experience.

## 3. Results

The international online convenience sample included 1,534 adults identifying as having ME and/or CFS who completed the questionnaire (347 additional respondents had incomplete surveys and were not included in this analysis). Respondents were from over 35 countries. As indicated in Table 1, 41.1% of participants reported currently living in the United States. The sample consisted of mostly females (84.6%). The majority of participants were white/Caucasian (97.5%), and 2% identified as being of Latino or Hispanic origin. Just over half of the participants were married or living with a partner (56.6%), 39.3% had a standard college degree, and 45.7% were receiving disability payments.

Table 1 indicates that 50.7% of participants had a diagnosis of CFS, 22.0% had a diagnosis of ME, and 27.2% had a diagnosis of both ME and CFS. For our entire sample, 94.4% reported being diagnosed by a medical doctor.

Descriptive statistics of PEM onset are reported in Table 2. Over half of participants had experienced onset of symptom exacerbation immediately after exertion (72.3%), while 91.4% had experienced delayed onset after exertion. To determine the length of the delay between exertion and the onset of PEM, participants selected periods for when the onset of PEM might occur when onset is delayed. A delay of between 1–2 days after exertion was experienced by 53.1% of the participants.

Table 3 describes PEM triggers, with 78.2% endorsing “basic activities of daily living”, 64.5% endorsing “positional changes”, and 93.2% endorsing “emotional stress (good or bad)”. Additionally, 84.9% said there were some instances in which the specific precipitants could not be identified. The highest endorsed non-exertion triggers reported by participants were as follows: emotional events (88.3%), noise (85.5%), and sensory overload (83.6%).

Table 4 reports the proportion of participants who endorsed the worsening of symptoms due to physical or cognitive exertion. The most commonly endorsed symptoms were as follows: reduced stamina and/or functional capacity (99.4%), physical fatigue (98.9%), cognitive exhaustion (97.4%), problems thinking (97.4%), unrefreshing sleep (95.0%), muscle pain (87.9%), insomnia (87.3%), muscle weakness/instability (87.3%), temperature dysregulation (86.9%), and flu-like symptoms (86.6%). The symptoms endorsed by less than half of the sample included the following: loss of appetite (49.0%), migraines (46.2%), cardiac pain and/or arrhythmia (41.2%), brain twangs (29.9%), burning sensation all over your skin (29.7%), paralysis/inability to move (29.4%), pre-menstrual symptoms (21.1%), and decreased heart rate (15.1%).

In order to gauge participant’s general experiences of PEM, participants were asked if they experienced any of the common phrases used to describe PEM (listed in Table 5) after exertion. All of the phrases were endorsed by over 90% of the sample.

The findings reported in Table 6 indicate that over half the participants (58.0%) said PEM lasts on average 3–6 days, with 1–2 days (38.9%), 1 week–1 month (46.7%), and 1–6 months (30.3%) also being frequently reported. Additionally, 67.1% of the sample had experienced a “crash” that never resolved. Over half of the sample (57.2%) said they had experienced an adrenaline surge during or after going beyond their energy limits, and the most commonly reported length of time was “a few hours” (35.8%). Further information about the natural history of participants’ ME/CFS illness are also provided in Table 6. The majority of subjects have been sick for over 10 years, with 97.1% reporting their illness being present for more than 50% of the time. Additionally, nearly half of participants described the course of their illness as fluctuating, experiencing good periods and bad periods. Lastly, nearly half of participants classified their status as being able to do light house work, but not being able to work part-time.

Table 6 also contains information on how participants were currently managing their PEM symptoms. Only 6% of patients with ME or CFS felt that pacing completely allowed them to avoid PEM, while the majority reported pacing only being effective some of the time and only at a moderate/mild level. Participants also identified the pacing method they used (e.g., 87.1% indicated it was based on their bodies’ reactions whereas 10.7% indicated it was with a heart rate monitor, and 17.3% indicated both).

Patients were also asked about tests to assess their cardiovascular health difficulties and orthostatic intolerance, which are common symptoms of ME and CFS and are often made worse after exertion. Almost a quarter (24.5%) indicated they had undergone a cardiopulmonary test and 29.7% indicated they had taken part in a stand lean/tilt table test. Of those patients, 9.3% had normal cardiopulmonary results, whereas 14.9% had abnormal results. Only 4.8% of the sample had completed an exercise test on back-to-back days.

At the end of the questionnaire, participants were asked if they felt this survey accurately captured their experiences of PEM, and 29.8% felt the survey was very accurate, 57.7% reported it was accurate, 10.7% were neutral, 1.2% thought it was not accurate, and 0.1% said it was not at all accurate.

## 4. Discussion

The objective of this study was to use community-based participatory research in an effort to develop a comprehensive way to assess PEM. Based on the comments and items suggested from patients, the following specific aspects of PEM were found to be the most critical domains: the timing of PEM onset, triggers of PEM, symptoms that are exacerbated following exertion or exposure to triggers, phrases used to describe consequences of PEM, duration of PEM, relationship between exertion and length of recovery, and the importance of considering personal characteristics (e.g., how long the patient has had ME/CFS, the course of their illness, their level of functioning, and coping methods used). The patient perspective provided the authors with the critical information to develop this survey of PEM. Of the patients who took part, 87.5% felt that the resulting survey was either very accurate or accurate.

Onset of symptom exacerbation after exertion was found to vary between patients. As shown in Table 2, the majority of patients experienced both immediate and delayed onset of PEM, and the extent of the delay of symptoms varied considerably. In addition to the unpredictability of PEM onset, several factors affect the duration of PEM before recovery, including the type, intensity, frequency, and duration of the exertion (see Table 6). These findings are consistent with patients’ reporting of prolonged recovery from PEM symptoms. In one study in which patients and healthy controls participated in a fatiguing exercise test, the patient group’s recovery was prolonged [21]. In addition, VanNess et al. [13] found patients with CFS, in comparison to healthy controls, take considerably longer to recover after completing a maximal cardiopulmonary exercise test the next day and a week later. Our findings are also consistent with a study by Chu et al. [11] who found that when comparing PEM symptom onset between those with ME or CFS to healthy controls, 87‒95% of controls had recovered within 24 h after completing an exercise test. Among those with ME and CFS, PEM symptoms peaked at 24 to 48 h later, and 45‒60% still experienced symptoms up to 5 days later.

Our survey also assessed specific triggers that bring on symptom exacerbation. The effects of physical and cognitive exertion on PEM have been well-established [13,21,22,23] and these findings are consistent with the current study. For example, only 37% of subjects reported being able to exercise “a little” without PEM-related symptoms, as long as they stay within “certain limits” (see Table 3). Furthermore, basic activities of daily living (e.g., getting dressed, cooking a meal, bathing), positional changes (e.g., going from lying down to standing up), and emotional stress lead to exacerbation in 78.2%, 64.5%, and 93.2% of patients, respectively.

Another issue explored involved whether there are precipitants of PEM beyond physical or cognitive exertion. The highest reported triggers in addition to physical/cognitive exertion were emotional events (88.3%), noise (85.3%), and sensory (83.6%) and visual overload (79.7%). This is consistent with past literature reporting these types of stimuli as exacerbating symptoms [24]. It has also been hypothesized that exposure to mold could trigger illness onset and PEM symptomology [25]. In our sample, 39.4% reported mold triggering their PEM. This is consistent with findings by Brewer, Thrasher, Straus, Madison, and Hooper [26], where 30% of patients with ME and CFS were reported to have multiple mycotoxins present in their bodies.

Partly as a function of this survey and the interactions with the patient community, there have been several additional developments in the assessment of PEM. First, Cotler et al. [27] found that use of the 5 recommended PEM DSQ items was an excellent screen in identifying PEM in patients with ME and CFS. In addition, as a second step in the process of assessing PEM, 5 additional DSQ items (including the assessment of duration of symptoms) were successfully used to differentiate PEM from other chronic illnesses. In addition, the findings from the patient survey reported on in this article were revised in order to construct a briefer, more concise measure of PEM, which was significantly related to physical functioning [28].

There are several limitations to this study. First, we did not obtain confirmation of ME or CFS diagnoses by independent medical personnel. In addition, we do not know what case definitions, if any, were used in their diagnoses. In addition, consistent with other ME and CFS studies, the sample was not demographically diverse. However, having a sample from several geographic regions did increase the generalizability of findings. Another limitation of the study was the length of the questionnaire. Though participants were presented with the option of pausing, it is reasonable that some may have still found it difficult to complete.

The open, participatory nature of this study provided a unique way of both designing the survey and gathering comprehensive information from the ME and CFS community regarding PEM. There are unique benefits that can accrue to the research and patient community by actively collaborating on instrument development as well as other policy issues, such as the selection of a name for the illness as well as the case definition [29]. By collaborating with the ME and CFS community, we have provided a model of community-based participatory research, which has multiple advantages to both the patient and research communities [30]. We close with this quote regarding what needs to occur to further this type of collaborative research in the ME and CFS areas:

“An alternative vision is still possible if those in power are willing to bring all interested parties to the table, including international representatives, historians on the science of illness criteria, and social scientists adept at developing consensus. In a collaborative, open, interactive, and inclusive process, issues may be explored, committees may be charged with making recommendations, and key gatekeepers may work collaboratively and transparently to build a consensus for change. Involve all parties—patients, scientists, clinicians, and government officials—in the decision-making process [31].”

## Figures and Tables

**Table 1 diagnostics-09-00026-t001:** Demographic characteristics of patients with myalgic encephalomyelitis (ME) and chronic fatigue syndrome (CFS) (*N* = 1534).

Age	M (SD)
	51.26 (13.08)
**Gender**	**% (n)**
Female	84.6 (1,298)
Male	14.9 (229)
**Race**	**% (n)**
White/Caucasian	97.5 (1,495)
Black/African American	0.3 (4)
American Indian or Alaska Native	0.7 (11)
Asian or Pacific Islander	1.1 (17)
Latino/Hispanic Origin	2.0 (30)
Prefer not to respond	1.4 (22)
**Marital Status**	**% (n)**
Married or living with partner	56.6 (869)
Never married	23.3 (357)
Divorced	13.9 (213)
Separated	2.6 (40)
Widowed	2.0 (31)
Prefer not to answer	1.2 (19)
**Education Level**	**% (n)**
Graduate/professional degree	29.1 (446)
Standard college/university degree	39.3 (603)
Partial college	22.1 (339)
High school or General Education Development (GED)	5.9 (91)
Some high school	2.5 (39)
Less than high school	0.8 (12)
**Employment Status**	**% (n)**
On disability	45.7 (701)
Working full-time	6.8 (104)
Working part-time	13.2 (203)
Homemaker	7.3 (112)
Student	3.3 (50)
Retired	18.1 (278)
Unemployed	16.0 (245)
Prior to leaving the workforce, did you cut back either in number of hours worked or in responsibilities	57.5 (880)
**Diagnosis**	**% (n)**
CFS	50.7 (777)
ME	22.0 (338)
Both ME and CFS	27.2 (418)
**Who diagnosed you?**	**% (n)**
Medical doctor	94.4 (1448)
Was the medical doctor an expert/knowledgeable of ME/CFS?	55.6 (853)
Alternative practitioner	5.5 (85)
Self-diagnosed	7.6 (117)
**Current Annual Income (in US dollars)**	**% (n)**
Less than $24,999	52.2 (801)
$25,000 to $49,999	14.7 (225)
$50,000 to $99,999	8.3 (128)
$100,000 to $149,999	2.8 (43)
$150,000 to $199,999	0.9 (14)
$200,000 to $249,999	0.2 (3)
$250,000 or more	1.0 (16)
Prefer Not to Respond	18.1 (277)
**Annual Income prior to becoming ill (in US dollars)**	**% (n)**
Less than $24,999	15.4 (237)
$25,000 to $49,999	25.0 (384)
$50,000 to $99,999	25.4 (390)
$100,000 to $149,999	6.7 (103)
$150,000 to $199,999	1.8 (27)
$200,000 to $249,999	1.2 (18)
$250,000 or more	1.7 (26)
Prefer Not to Respond	19.9 (305)

Note: Percentages may not add up to 100% due to missing data. For employment status, there were also several open response questions asking about what conditions participants received disability for, and for current and past job titles.

**Table 2 diagnostics-09-00026-t002:** Onset (*N* = 1534).

Items	% (n)
**Immediate onset of symptom exacerbation**	**72.3 (1109)**
All the time	9.9 (152)
Most of the time	21.9 (336)
About half the time	24.1 (369)
A little of the time	15.6 (239)
**Delayed onset of symptom exacerbation**	**91.4 (1402)**
All the time	21.8 (335)
Most of the time	37.1 (569)
About half the time	23.4 (359)
A little of the time	8.1 (125)
**How long after the exertion does your symptom exacerbation occur ***	
1 h or less	16.5 (253)
2–6 h	33.1 (508)
7–12 h	31.0 (476)
13–24 h	43.2 (662)
1–2 days	53.1 (815)
3–4 days	15.7 (241)
5–6 days	4.5 (69)
More than 1 week	4.2 (65)

Note: * For this item, participants could select more than one answer. There is also an option for participants to describe what activities and which symptoms affect immediate and/or delayed onset.

**Table 3 diagnostics-09-00026-t003:** Triggers (*N* = 1534).

Items	% (n)
**Basic activities of daily living trigger symptom exacerbation**	**78.2 (1199)**
All of the time	20.8 (319)
Most of the time	24.1 (370)
About half the time	17.7 (272)
A little of the time	15.3 (234)
**Positional changes lead to symptom exacerbation**	**64.5 (990)**
All of the time	14.9 (229)
Most of the time	20.0 (307)
About half the time	15.5 (238)
A little of the time	13.9 (213)
**Emotional stress (good or bad) lead to symptom exacerbation**	**93.2 (1429)**
All of the time	34.0 (522)
Most of the time	29.2 (448)
About half the time	18.3 (280)
A little of the time	11.5 (177)
Instances in which the specific precipitants cannot be identified	84.9 (1302)
Able to exercise a little as long as you stay within certain limits without symptom exacerbation	37.0 (567)
Takes less exposure than usual to trigger PEM on days you are recovering from symptom exacerbation	94.3 (1447)
Sensitized to particular triggers so they cause an even more abnormal response over time	48.1 (738)
Severity of the PEM reaction proportionate to how far beyond your limits you have gone	80.9 (1241)
Mild overexertion over several days produces an abnormal physical or cognitive response	96.8 (1485)
Multiple occurrences of PEM that cause your overall health status to become worse over weeks/months	84.4 (1295)
Intolerance to stimulation causes worsening in symptoms, but is not prolonged if stimulus is removed	79.5 (1219)
Fighting off an infection (flu, cold, bladder infection) causes a worsening in all/most of your symptoms	82.3 (1262)
Length of time for recovery correlates with the severity of PEM	79.6 (1221)
**Do you have other triggers such as**	
Emotional events (good or bad)	88.3 (1354)
Noise	85.3 (1308)
Sensory overload	83.6 (1282)
Visual overload	79.7 (1223)
Heat	74.4 (1141)
Light	68.8 (1055)
Cold	66.3 (1017)
Foods	61.0 (935)
Chemicals	58.0 (889)
Watching movement (such as watching a video)	52.5 (806)
Vibration	47.1 (722)
Drugs used for medication	47.4 (727)
Mold	39.4 (605)
Supplements	27.4 (420)

**Table 4 diagnostics-09-00026-t004:** Symptoms made worse due to physical or cognitive exertion (*N* = 1534).

Items	% (n) “Yes”	% (n) at “2” Threshold	Mean (SD)
1. Reduced stamina and/or functional capacity	99.4 (1525)	98.0 (1504)	90.60 (17.16)
2. Physical fatigue	98.9 (1517)	98.3 (1508)	87.53 (18.26)
3. Cognitive exhaustion	97.4 (1494)	92.0 (1412)	77.64 (24.87)
4. Problems thinking	97.4 (1494)	92.6 (1420)	78.47 (24.87)
5. Unrefreshing sleep	95.0 (1457)	91.1 (1398)	80.57 (27.65)
6. Muscle pain	87.9 (1349)	81.5 (1250)	69.41 (33.95)
7. Insomnia	87.3 (1339)	75.1 (1152)	62.40 (34.30)
8. Muscle weakness/instability	87.3 (1339)	77.2 (1185)	64.03 (33.86)
9. Temperature dysregulation	86.9 (1333)	75.2 (1153)	63.76 (34.75)
10. Flu-like symptoms	86.6 (1329)	74.4 (1142)	59.52 (33.43)
11. Aches all over your body	85.6 (1313)	79.5 (1219)	68.68 (35.58)
12. Physically fatigued while mentally wired	82.1 (1259)	72.8 (1116)	59.00 (35.65)
13. Dizziness	80.7 (1238)	56.0 (859)	46.28 (33.19)
14. Gastro-intestinal problems	78.6 (1206)	59.3 (910)	49.90 (36.02)
15. Headaches	78.0 (1197)	56.5 (866)	46.48 (34.52)
16. Ataxia	77.6 (1191)	57.8 (886)	47.62 (35.18)
17. Increased heart rate/heart palpitations	77.4 (118)	64.9 (996)	52.28 (36.51)
18. Weak or stiff neck	74.6 (1144)	61.0 (936)	51.35 (38.20)
19. Joint pain	73.0 (1120)	59.5 (912)	49.17 (37.86)
20. Problems with speech	72.4 (1110)	50.0 (767)	40.22 (33.14)
21. Sore throats	70.9 (1087)	47.2 (724)	38.92 (33.55)
22. Muscle twitching	68.1 (1045)	40.9 (627)	35.12 (32.38)
23. Night sweats and chills	67.7 (1038)	46.9 (720)	38.48 (34.69)
24. Sore eyes	67.0 (1028)	49.0 (752)	39.91 (35.70)
25. Nerve pain	63.3 (971)	48.8 (748)	40.65 (38.16)
26. Sore lymph nodes	62.9 (965)	44.0 (675)	36.36 (35.28)
27. Nausea	62.2 (954)	38.1 (584)	31.89 (32.13)
28. Tinnitus	60.3 (925)	39.8 (611)	37.42 (38.96)
29. Trouble breathing	57.8 (887)	40.9 (628)	33.97 (35.67)
30. Neurological symptoms	57.0 (875)	42.8 (656)	34.60 (36.14)
31. Excessive sleep	54.4 (835)	44.5 (682)	36.23 (38.58)
32. Loss of appetite	49.0 (752)	30.9 (474)	25.41 (31.62)
33. Migraines	46.2 (708)	24.6 (378)	21.92 (29.27)
34. Cardiac pain and/or arrhythmia	41.2 (632)	24.8 (381)	21.12 (30.30)
35. Brain twangs	29.9 (459)	17.7 (272)	15.00 (26.82)
36. Severe burning sensation all over skin	29.7 (456)	18.3 (280)	15.96 (28.87)
37. Paralysis/inability to move	29.4 (451)	9.4 (144)	11.49 (21.91)
38. Premenstrual symptoms	21.1 (323)	16.4 (251)	13.56 (29.25)
39. Decreased heart rate	15.1 (231)	7.4 (114)	6.88 (19.09)

Note: % endorsed “yes” means they responded yes to experiencing symptom at any level. % endorsed at “2” threshold means that they experience the symptom at least half the time. Means reflect frequency only (0–100 scale).

**Table 5 diagnostics-09-00026-t005:** If you go beyond your energy limits by engaging in pre-illness tolerated exercise or activities of daily living, do you experience any of the following? (*N* = 1534).

Items	% (n)
An onset that is immediate or delayed by hours or days	98.5 (1511)
Post-exertional exhaustion	98.3 (1508)
A loss of functional capacity and/or stamina	98.2 (1506)
Symptom exacerbation	98.1 (1505)
A severity and duration of symptoms that are out of proportion to the initial trigger	97.4 (1494)
An abnormal response to minimal amounts of physical and/or cognitive exertion	97.3 (1492)
Substantial reduction in pre-illness activity level	96.9 (1486)
A prolonged recovery that can last days, weeks, or months	96.2 (1475)
Global worsening of multi-systemic symptoms	94.0 (1442)
Prolonged worsening of symptoms	92.9 (1425)

**Table 6 diagnostics-09-00026-t006:** Duration of PEM, illness course, and functioning (*N* = 1534).

Items	% (n)
**Length of prolonged, unpredictable recovery period**	**95.2 (1460)**
Within 24 h	14.1 (216)
Between 1 and 2 days	38.9 (596)
Between 3 and 6 days	58.0 (890)
Between 1 week and 1 month	46.7 (717)
Between 1 and 6 months	30.3 (465)
Between 6 months and 1 year	13.6 (209)
Between 1 and 2 years	9.8 (151)
Over 2 years	12.3 (189)
Crash that has never resolved	67.1 (1029)
**Severity and duration out-of-proportion to the TYPE of exertion**	**96.0 (1473)**
All of the time	59.0 (905)
Most of the time	26.1 (401)
About half the time	8.7 (133)
A little of the time	2.0 (31)
**Severity and duration out-of-proportion to the INTENSITY of exertion**	**94.8 (1454)**
All of the time	59.5 (913)
Most of the time	26.5 (406)
About half the time	6.6 (102)
A little of the time	1.9 (29)
**Severity and duration out-of-proportion to the DURATION of exertion**	**90.4 (1386)**
All of the time	56.9 (873)
Most of the time	25.6 (393)
About half the time	5.1 (78)
A little of the time	1.8 (28)
**Severity and duration out-of-proportion to the FREQUENCY of exertion**	**84.9 (1302)**
All of the time	51.5 (790)
Most of the time	24.8 (380)
About half the time	5.5 (85)
A little of the time	2.6 (40)
**Adrenaline surges during or after going beyond energy limit**	**57.2 (878)**
**Length of adrenaline surge before crashing ***	
A few minutes	13.0 (200)
A few hours	35.8 (549)
About 24 h	16.5 (253)
Less than a week	6.1 (94)
About 1 week	1.3 (20)
Over 1 week	1.3 (20)
**How long ago did your problem with ME/CFS begin?**	
6–11 months ago	0.6 (9)
1–2 years ago	2.9 (45)
3–5 years ago	12.1 (186)
6–10 years ago	15.9 (244)
Over 10 years ago	53.7 (823)
Since childhood/adolescence	14.8 (227)
**Has your illness been present for more than 50% of the time since you became ill?**	**97.1 (1489)**
**How would you describe the course of your illness?**	
Constantly getting worse	29.3 (450)
Constantly improving	1.4 (22)
Persisting (no change)	15.4 (237)
Relapsing and remitting	7.4 (113)
Fluctuating	46.2 (708)
**Which statement best describes your illness over the last 6 months?**	
I can do all work or family responsibilities without any problems with my energy	0.1 (2)
I can work full-time/finish some family responsibilities, but I have no energy left	2.6 (40)
I can work full-time, but I have no energy left for anything else	4.6 (71)
I can only work part-time at work or on some family responsibilities	14.9 (228)
I can do light housework, but I cannot work part-time	43.0 (659)
I can walk around the house, but I cannot do light housework	29.9 (459)
I am not able to work or do anything, I am bedridden/completely incapacitated	4.8 (73)
Pacing allows me to completely avoid symptom exacerbation	6.0 (92)
Pacing allows me to avoid symptom exacerbation only to a certain degree	87.7 (1345)
**How frequently is pacing effective?**	
All the time	2.3 (35)
Most of the time	22.8 (350)
About half the time	34.1 (523)
A little of the time	27.8 (427)
**How effective is pacing in reducing the level of severity of symptoms?**	
Very effective	7.6 (117)
Moderately effective	37.2 (570)
Mildly effective	34.2 (525)
Barely effective	8.2 (126)
**If you are pacing, is it:**	
Based on body symptoms and reactions to triggers	87.1 (1336)
With a heart rate monitor	10.7 (164)
Both of the above	17.3 (265)

Note: * For these items, participants could select more than one answer. There is also an option for participants to describe pacing techniques not listed.

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
