# Peer review of "Assessment of Post-Exertional Malaise (PEM) in Patients with Myalgic Encephalomyelitis (ME) and Chronic Fatigue Syndrome (CFS): A Patient-Driven Survey"

_diagnostics, 2019, doi:10.3390/diagnostics9010026_

Round 1
Reviewer 1 Report
I have two major comments: 1) Change the title to "Patient driven survey" to "Patient survey" Lines 94 to 110; There is not enough to justify this was a participatory, action-based approach. I suggest this part is omitted from the text, as the paper represents a survey with people with "patients", but we do not see anything on true participatory approach. Alternatively, the authors should explore in some details how participation happened. The text that should stay there is "!The aim of this [paper was to report the preliminary descriptive results of survey of people with self-reported people with me/cfs. 2) The numbers and percentages on tables not always match and they must be reviewed.
Author Response
Point 1: Change the title to "Patient driven survey" to "Patient survey" Lines 94 to 110; There is not enough to justify this was a participatory, action-based approach. I suggest this part is omitted from the text, as the paper represents a survey with people with "patients", but we do not see anything on true participatory approach. Alternatively, the authors should explore in some details how participation happened. The text that should stay there is "The aim of this [paper was to report the preliminary descriptive results of survey of people with self-reported people with me/cfs.
Response 1: It is important to the authors that we acknowledge the collaboration between researchers and the patient community because without the active participation of the patients, this study and the development of the new PEM questionnaire would have never come to fruition. We have revised the introduction, methods, and discussion sections to further explain how this study is an example of participatory research. We include a more detailed history of participatory approaches in research, and specifically in the field of ME and CFS to support our position.
Point 2: The numbers and percentages on tables not always match and they must be reviewed.
Response 2: All numbers and percentages on tables have been reviewed and are correct.
Reviewer 2 Report
This is an excellent and well-presented study which captures the complexity of the patient experience of post-extertional malaise (PEM) in chronic fatigue syndrome (CFS) and myalgic encephalomyelitis (ME).
Author Response
Thank you for taking the time to review this article and for your feedback.
Reviewer 3 Report
This is in my opinion a useful contribution to an important subject
PEM is the cardinal feature of ME and has not received enough attention
Author Response

(The authors gave the same response as above.)
